# Frailty and depression predict instrumental activities of daily living in older adults: A population-based longitudinal study using the CARE75+ cohort

**Peter A. Coventry**[1]*, **Dean McMillan**[2], **Andrew Clegg**[3], **Lesley Brown**[4], **Christina van der Feltz-Cornelis**[2], **Simon Gilbody**[2], **Shehzad Ali**[1,5]

1 Department of Health Sciences, University of York, York, United Kingdom, 2 Department of Health Sciences and Hull York Medical School, University of York, York, United Kingdom, 3 Academic Unit of Elderly Care and Rehabilitation, University of Leeds, Bradford, United Kingdom, 4 Academic Unit of Elderly Care and Rehabilitation, Bradford, Institute for Health Research, Bradford, United Kingdom, 5 Department of Epidemiology and Biostatistics, Western University, London, Ontario, Canada

* peter.coventry@york.ac.uk

**Data Availability Statement:** The ethical approvals and governance permissions for CARE 75+ cohort (Bradford and Leeds Research Ethics Committee

## Abstract

### Objectives

To evaluate if depression contributes, independently and/or in interaction with frailty, to loss of independence in instrumental activities of daily living (ADL) in older adults with frailty.

### Methods

Longitudinal cohort study of people aged ≥75 years living in the community. We used multi-level linear regression model to quantify the relationship between depression (≥5 Geriatric Depression Scale) and frailty (electronic frailty index), and instrumental activities of daily living (Nottingham Extended Activities of Daily Living scale; range: 0–66; higher score implies greater independence). The model was adjusted for known confounders (age; gender; ethnicity; education; living situation; medical comorbidity).

### Results

553 participants were included at baseline; 53% were female with a mean age of 81 (5.0 SD) years. Depression and frailty (moderate and severe levels) were independently associated with reduced instrumental activities of daily living scores. In the adjusted analysis, the regression coefficient was -6.4 (95% CI: -8.3 to -4.5, p<0.05) for depression, -1.5 (95% CI: -3.8 to 0.9, p = 0.22) for mild frailty, -6.1 (95% CI: -8.6 to -3.6, p<0.05) for moderate frailty, and -10.1 (95% CI: -13.5 to -6.8, p<0.05) for severe frailty. Moreover, depression interacted with frailty to further reduce instrumental activities of daily living score in individuals with mild or moderate frailty. These relationships remained significant after adjusting for confounders.

### Conclusion

Frailty and depression are independently associated with reduced independence in instrumental activities of daily living. Also, depression interacts with frailty to further reduce

study [ref: 14/YH/1120]) do not allow for raw data to be deposited in a repository. However data enquiries for accessing Care 75+ can be made to the Yorkshire and the Humber – Bradford Leeds NHS Health Research Authority via bradfordleeds. rec@hra.nhs.uk.

**Funding:** The research was funded by the NIHR Applied Research Collaboration Yorkshire and Humber https://www.arc-yh.nihr.ac.uk/. The views expressed are those of the author(s), and not necessarily those of the NIHR or the Department of Health and Social Care.

**Competing interests:** The authors have declared that no competing interests exist.

independence for mild to moderately frail individuals, suggesting that clinical management of frailty should integrate physical and mental health care.

## Introduction

Globally, populations are rapidly ageing owing to increase in life expectancy and falling fertility rates. The fastest growing population is the oldest old, those aged 85 years and over. In the UK, the population of those aged ≥85 years is set to double to 3.2 million or 4% of the total population by mid-2041, and treble by 2066 with profound implications for health and social care services [1]. The likelihood of disability and/or experiencing multiple long term conditions, so-called multimorbidity, increases with age among those aged ≥65 years, leading to the possibility of increased time spent in poor health [2]. While there is some evidence that morbidity has become compressed into later life, in high income countries 50% of disease burden in those aged ≥60 years is attributable to long term conditions and there is considerable uncertainty about the health of future generations of older adults [3].

Using disease based approaches to understanding and managing the complex health needs of older adults has limited value and using frailty to characterise health status in older adults has emerged as a worthwhile alternative approach. There are a number of frailty definitions but there is broad consensus that it is characterised by loss of biological reserves, failure of homeostatic mechanisms and heightened vulnerability to adverse health outcomes, including falls, hospitalisations and mortality [4]. Frailty is common in later life and the risk of frailty increases substantially with age, affecting 1 in 10 people aged ≥65 years and between 25% and 50% of adults aged ≥85 years. The urgency to address this growing problem is reflected in UK health policy which has highlighted the need to develop and implement innovative care models to support older people with frailty [5].

A recent systematic review with meta-analysis (48 studies; n = 78122 participants) demonstrated that multimorbidity increased the risk of frailty two fold and in pooled analysis multimorbidity was significantly associated with frailty in community-dwelling people (OR 2.27; 95%CI 1.97–2.62) [6]. However, while most people with frailty will have multimorbidity, not all people with multimorbidity will become frail. While physical and biological factors are pre-eminent in explaining vulnerability to poor health among those with frailty, mental health is also implicated in adverse outcomes in older adults with disabling long-term conditions [7, 8]. Depression is longitudinally associated with frailty and those with frailty have a four-fold increased odds of having depression [9, 10]. Likewise, those with depression have similar odds of having frailty, pointing to the reciprocal nature between depressive states and frailty. Furthermore, the clinical manifestations of depression overlap with some of the key phenotypic components of frailty, such as exhaustion, low energy expenditure, weight loss and possibly slow walking speed owing to lack of motivation. Indeed there is a strong argument to consider depression and frailty as distinct but over-lapping constructs and as such these two inter-related syndromes should be evaluated together in efforts to understand how to optimally manage health and reduce disability in older adults [11].

Frailty predicts functioning and disability in activities of daily living (ADL) in community dwelling older adults and functioning and disability are better markers of survival and future health outcomes than disease status and comorbidity [12, 13]. However, it is not known if depression contributes either independently or in combination with frailty and multimorbidity to disability in older adults with frailty. IADL represent functional competence in everyday higher level tasks such as shopping or preparing a meal and are known to decline before basic

ADLs associated with self-care. In this sense IADLs are appropriate markers of loss of independence and disability in older adults with a broad range of frailty. We therefore evaluated whether depression predicts instrumental ADL (IADL) disability in a unique population-based cohort of adults aged ≥75 years with a focus on frailty status and frailty trajectories.

## Materials and methods

### Ethical approval

The Bradford and Leeds Research Ethics Committee granted ethical approval for the CARE 75 + study (ref: 14/YH/1120). The CARE 75+ study is registered at ISRCTN16588124.

### Study population

The Community Ageing Research 75+ Study (CARE75+) (Trial registration number ISRCTN16588124) is an on-going, longitudinal population based cohort study of older people. The data that comprise the analytic sample in this study were collected between January 2014 and December 2018. Community dwelling older people aged ≥75 years were eligible for inclusion. Exclusion criteria were: care home residents at point of recruitment; bedbound at home; have terminal cancer; in receipt of the Amber Care Bundle and estimated life expectancy of three months or less; and in receipt of palliative care services. Potential participants were recruited via their general practices. Participants undergo a range of cognitive, physical and psychosocial assessments at baseline, 6 and 12 months. Assessment were conducted face-to-face in the person's home, with an optional modified follow-up at 6 months via telephone or the internet. Full details of the CARE75+ protocol including recruitment, consent and data collection, entry, coding, security and storage have previously been reported [14]. The protection and security of CARE75+ data was ensured through an information sharing agreement between Bradford Teaching Hospitals NHS Foundation Trust and the University of York in accordance with the Data Protection Act 1998.

### Outcomes and predictors

The primary outcome was disability in IADL, measured using the Nottingham Extended ADL (NEADL) scale [15]. The NEADL scale was first validated in stroke patients and includes sub-scales for mobility, kitchen and domestic ability, and leisure activity. It is scored between 0 and 66 with higher scores indicating greater independence and less disability.

Our list of potential predictors were based on the frailty literature: age, gender, ethnicity (white; mixed white/black Caribbean; black Caribbean; Asian Pakistani; Asian Indian; Asian Bangladeshi; other), education, self-reported medical comorbidities (using Katz index). Depression was measured using the 15-item Geriatric Depression Scale (GDS) Short-Form [16]. The GDS-15 is scored from 0 to 15, with a score of ≥5 is suggestive of depressive symptoms and a score ≥ 10 is almost always indicative of depression. In meta-analysis the GDS-15 has an average sensitivity of 0.805 and specificity of 0.750 for identifying cases of major depressive disorder [17]. Frailty was measured using the electronic frailty index (eFI) which is based on the cumulative deficit model of frailty [18]. The eFI includes 36 equally weighted deficit variables based on Read codes and recorded routinely in the primary care electronic health record. The eFI score is derived from the number of deficits present relative to the proportion of the total possible and identifies four frailty categories: 0–0.12 = fit; 0.13–0.24 = mild frailty; 0.25–0.36 = moderate frailty; >0.36 = severe frailty. The eFI has good convergent validity with other research standard frailty measures [14]. We also explored if comorbidity predicted instrumental ADL given their strong association with functional decline in community

dwelling older adults [19, 20]. Comorbidity data were collected using the self-reported Katz comorbidity questionnaire that asks questions about the presence of absence of long term physical and mental conditions that are given a "yes" or "no" response, giving a total number of comorbidities [21]. Variables were measured at baseline, 6-months and 12-months [22].

### Statistical analysis

Regression analyses were conducted with NEADL score as a continuous dependent variable (range: 0–66, with higher score implying greater independence) and frailty levels (categorical: no, mild, moderate or severe frailty), depression status (binary variable, based on GDS-15 score of ≥5) and the interaction between frailty and depression. Other covariates included age, gender, ethnicity (white vs non-white), level of education (no education; GCSE or AS/A levels; Higher National Qualification (HNQ), diploma or University degree), living situation (living alone, with partner/spouse only, with family), comorbidity (measured using self-assessed Katz scale) and time (baseline, 6-months and 12-months). A multi-level regression model with linear mixed-effects was used which allows use of all available data at each time point. The model included individual-level random intercepts to account for repeated measurements nested within an individual (i.e. repeated observations nested within individual-level). Models were fitted with increasing complexity, starting first with main effects for frailty and depression and then introducing interaction effects. The interaction term evaluates whether the effect of frailty on instrumental daily activities of living is moderated by the level of depression, i.e. does depression influence the relationship between frailty and daily activities. Finally, based on this model, marginal (population-averaged) effects were estimated. The analysis was conducted in Stata v.15.1.

## Results

This study is reported in accordance with the Strengthening The Reporting of Observational Studies in Epidemiology (STROBE) checklist (S1 Checklist). From a total of 1875 potentially eligible participants we identified baseline data for 553 participants (Fig 1). We recorded a total of 14 deaths; half of these participants, i.e. 7/14, completed all three waves of the study (i.e. they died after the 12-month follow-up period). As a result, deaths accounted for only 7 cases of missing data during the study follow-up. We did not collect data on possible transition to a care home. Of the 553 participants included at baseline just over half (53%) were female with a mean age of 81 (5.0 SD) years. The majority (89%) were of white ethnicity and 57% had no formal educational qualifications. A high proportion were either living alone (41%) or living with a partner or spouse (44%). The mean baseline total score on the NEADL (50.95; SD = 15.47) indicated that participants generally had good global instrumental functioning; this score was lower at follow-up time points (6 months: mean = 46.19; SD = 19.31) and 12 months: mean = 46.27; SD = 18.42). All participants had multimorbidity with a mean of five long term health conditions. Fifty eight participants had GDS scores that were suggestive of depression (Table 1). Table 2 shows the number of people with depression at each level of frailty over time.

### Regression results for model 1: Unadjusted main effects analysis

Instrumental ADL, measured by the NEADL scale, was significantly lower at 6 and 12 months compared to the baseline. Both frailty and depression were independently and negatively associated with NEADL score, i.e. depressed and more frail individuals had lower levels of independence and higher level of disability (Table 3).

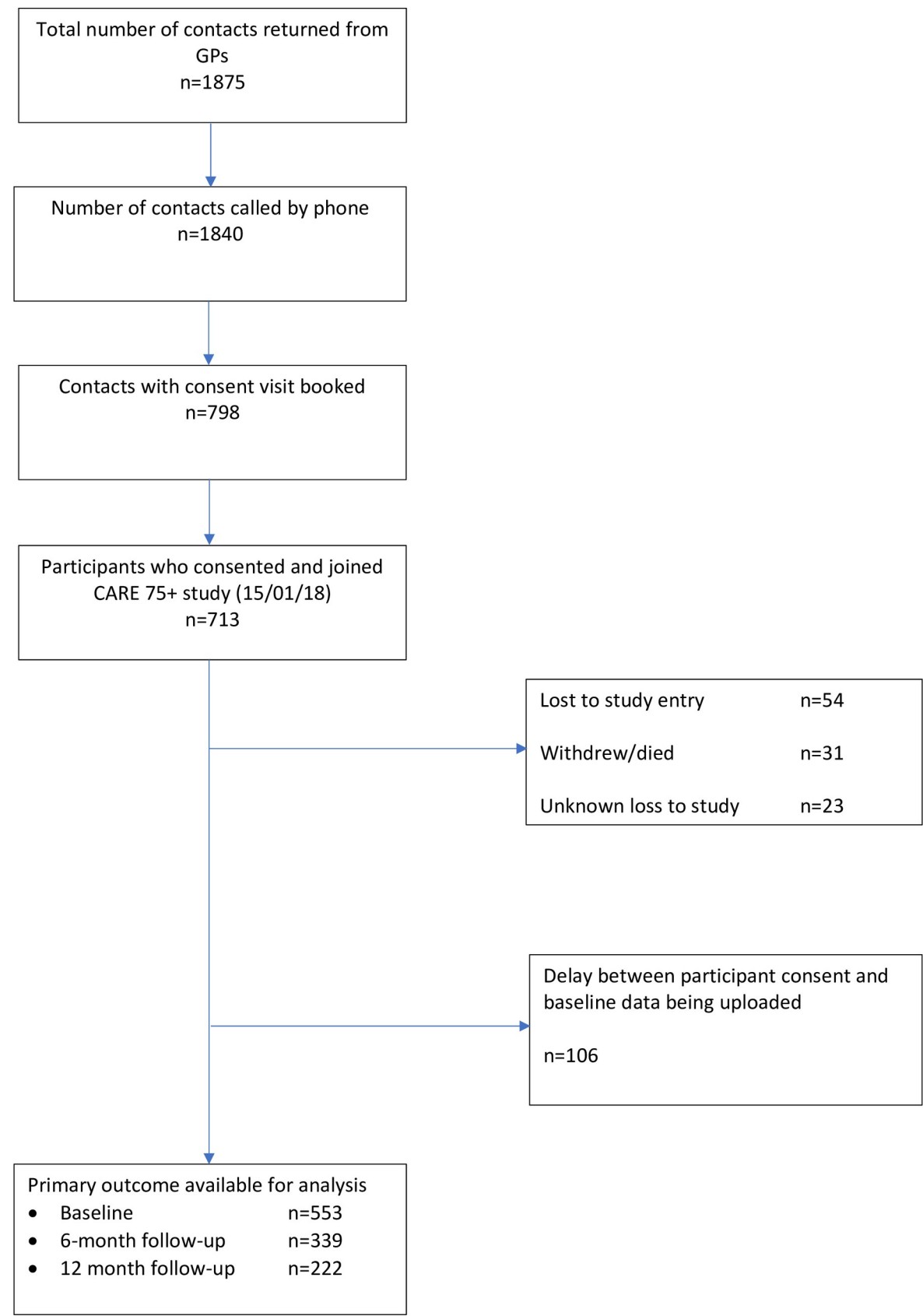

**Fig 1. STROBE flow diagram.**

## Regression results for model 2: Unadjusted main and interaction effects analysis

In model 2, an interaction term for depression and frailty was introduced. This model shows that there was a statistically significant interaction between depression and mild frailty (-14.2, 95% CI -25.5 to -2.8, p<0.05) and moderate frailty levels (-11.1, 95% CI -22.0 to -0.24, p<0.05) but not severe frailty. This implies that, in patients with mild or moderate frailty, depression is associated with even lower levels of independence in terms of instrumental activities of daily living.

## Regression results for model 3: Adjusted main effects analysis

Model 3 adjusted for potential confounders and found that older age, non-white ethnicity, lower education level, living situation, and higher comorbidity were all significantly associated

**Table 1. Characteristics of participants at baseline.**

|  | Mean (continuous variables); no. of observations (categorical variables) | SD (continuous variables); percentage (categorical variables) |
|---|---|---|
| Females, n (%) | 296 | 53.5 |
| Mean (SD) age, years | 81.0 | 5.05 |
| Ethnicity, n (%) |  |  |
| White | 494 | 89.49 |
| Asian Bangladeshi | 3 | 0.54 |
| Asian Indian | 3 | 0.54 |
| Asian Pakistani | 50 | 9.06 |
| Mixed white/black Caribbean | 1 | 0.18 |
| Black Caribbean | 1 | 0.18 |
| Missing | 1 | 0.18 |
| Education, n (%) |  |  |
| No qualifications | 316 | 57.2 |
| GCSE | 79 | 14.3 |
| AS and A levels | 17 | 3.0 |
| Higher National Certificate | 36 | 6.5 |
| Diploma | 35 | 6.3 |
| Bachelor's degree | 37 | 6.7 |
| Postgraduate | 15 | 2.7 |
| Missing | 17 | 3.0 |
| Living situation, n (%) |  |  |
| Alone | 225 | 40.8 |
| With family | 81 | 14.7 |
| With partner/spouse | 245 | 44.4 |
| Missing | 1 | 0.18 |
| Nottingham Extended ADL score (baseline), mean (SD) | 50.95 | 15.5 |
| Geriatric Depression Scale score (baseline), mean (SD) | 2.4 | 2.5 |
| Comorbidity (based on Katz comorbidity index), mean (SD) | 5.27 | 1.55 |

ADL: activities of daily living; AS/A Level: advanced subsidiary/advanced level; GCSE: general certificate of education; SD; standard deviation

**Table 2. Cross-tabulation of depression and frailty at baseline, 6 months, and 12 months.**

| Baseline | Electronic Frailty Index (EFI) | | | |
|---|---|---|---|---|
| | **No frailty** | **Mild frailty** | **Moderate frailty** | **Severe frailty** |
| Not depressed | 84 | 127 | 116 | 28 |
| Depressed | 0 | 15 | 30 | 20 |
| | Electronic Frailty Index (EFI) | | | |
| 6-months | **No frailty** | **Mild frailty** | **Moderate frailty** | **Severe frailty** |
| Not depressed | 47 | 79 | 74 | 21 |
| Depressed | 0 | 8 | 20 | 11 |
| | Electronic Frailty Index (EFI) | | | |
| 12-months | **No frailty** | **Mild frailty** | **Moderate frailty** | **Severe frailty** |
| Not depressed | 33 | 52 | 68 | 20 |
| Depressed | 2 | 2 | 10 | 8 |

with impaired instrumental ADL (Table 2). Coefficients for the association between frailty and instrumental ADL were smaller after adjusting for confounders but remained significant for all levels except mild frailty. Compared with no frailty, moderate and severe frailty levels were associated with about 6 and 10 points lower scores on NEADL, respectively. The coefficient for depression was also statistically significant and indicate that depressed individuals had 6.4 points (95% CI -8.274 to -4.549) lower NEADL score compared with non-depressed individuals).

## Regression results for model 4: Adjusted main effects and interaction analysis

Finally, model 4 evaluated interaction effect in the adjusted model (i.e. same as model 3 but with interaction terms). The results showed that after adjusting for potential confounders the interaction between mild frailty and depression was still statistically significant (-12.5, 95% CI -24.8 to -0.26, $p < 0.05$). The interaction between moderate frailty and depression was weakly significant (-11.2, 95% CI -23.1 to 0.71, $p < 0.1$). The interaction term was not significant for severe frailty (-7.3, 95% CI -19.5 to 5.0).

The results imply that individuals with mild or moderate frailty who also experience depressive symptoms are likely to have higher disability in instrumental ADL compared with non-depressed individuals with the same level of frailty. This relationship is represented in the margins plot which shows that the predicted NEADL score in depressed individuals is lower than non-depressed individuals (Fig 2).

## Discussion

Using a large population based cohort of older adults aged ≥75 years we showed that frailty status and depression independently predicted disability in instrumental ADL. More severe frailty and depressive symptoms predicted poorer instrumental ADL respectively. Beta-coefficients were slightly attenuated when frailty and depression were modelled together but both variables remained significant predictors even when adjusted for comorbidity and other known confounders. There was evidence that depression moderates the relationship between frailty and instrumental ADL, but only in those with mild and moderate frailty. The absence of an effect in those with more severe frailty might stem from an underpowered analysis owing to smaller numbers of people with severe frailty with above threshold depressive symptoms.

**Table 3. Results of linear multilevel models with NEADL score as the dependent variable and individual-level random intercepts.**

| VARIABLES | Unadjusted models | | | | Adjusted models | | | |
|---|---|---|---|---|---|---|---|---|
| | Model 1: unadjusted, no interaction | | Model 2: unadjusted, with interaction | | Model 3: adjusted, no interaction | | Model 4: adjusted, with interaction | |
| | Coefficients (95% CI) | p-value | Coefficients (95% CI) | p-value | Coefficients (95% CI) | p-value | Coefficients (95% CI) | p-value |
| **Time point** (reference: baseline) | | | | | | | | |
| 6-months | -1.017 | 0.063 | -0.991 | 0.068 | -0.966 | 0.078 | -0.961 | 0.078 |
| | (-2.087 to 0.053) | | (-2.053 to 0.071) | | (-2.040 to 0.107) | | (-2.028 to 0.107) | |
| 12-months | -1.163 | 0.053 | -1.289** | 0.032 | -1.282* | 0.033 | -1.392* | 0.020 |
| | (-2.343 to 0.017) | | (-2.466 to -0.112) | | (-2.461 to -0.103) | | (-2.568 to -0.215) | |
| **EFI: reference** (reference: fit) | | | | | | | | |
| EFI: mild frailty | -3.653** | 0.010 | -3.313* | 0.019 | -1.464 | 0.223 | -1.153 | 0.340 |
| | (-6.423 to -0.883) | | (-6.088 to -0.538) | | (-3.818 to 0.891) | | (-3.519 to 1.213) | |
| EFI: moderate frailty | -10.507** | <0.001 | -10.407** | <0.001 | -6.104** | <0.001 | -5.836** | <0.001 |
| | (-13.441 to -7.572) | | (-13.354 to -7.461) | | (-8.606 to -3.601) | | (-8.365 to -3.308) | |
| EFI: severe frailty | -17.290** | <0.001 | -18.058** | <0.001 | -10.138** | <0.001 | -11.053** | <0.001 |
| | (-20.946 to -13.634) | | (-21.924 to -14.192) | | (-13.525 to -6.751) | | (-14.610 to -7.496) | |
| **Depressed (based on GDS-15)** | -6.710** | <0.001 | 4.213 | 0.434 | -6.411** | <0.001 | 3.958 | 0.505 |
| | (-8.684 to -4.735) | | (-6.343 to 14.769) | | (-8.274 to -4.549) | | (-7.677 to 15.592) | |
| **Interaction: Frailty X Depression** | | | | | | | | |
| Interaction: mild frailty X depressed | | | -14.189** | 0.014 | | | -12.526** | 0.045 |
| | | | (-25.526 to -2.851) | | | | (-24.790 to -0.262) | |
| Interaction: moderate frailty X depressed | | | -11.143* | 0.045 | | | -11.200 | 0.065 |
| | | | (-22.042 to -0.244) | | | | (-23.115 to 0.714) | |
| Interaction: severe frailty X depressed | | | -8.469 | 0.144 | | | -7.281 | 0.244 |
| | | | (-19.829 to 2.892) | | | | (-19.536 to 4.974) | |
| **Age (in years)** | | | | | -0.703** | <0.001 | -0.713** | <0.001 |
| | | | | | (-0.899 to -0.507) | | (-0.909 to -0.517) | |
| **Female** | | | | | -0.634 | 0.517 | -0.772 | 0.430 |
| | | | | | (-2.550 to 1.283) | | (-2.688 to 1.145) | |
| **Ethnicity: white** | | | | | 15.274** | <0.001 | 15.315** | <0.001 |
| | | | | | (12.189 to 18.359) | | (12.233 to 18.396) | |
| **Education** (reference: no formal qualification) | | | | | | | | |
| GCSE or AS/A levels | | | | | 3.795** | 0.002 | 3.705** | 0.003 |
| | | | | | (1.351 to 6.240) | | (1.255 to 6.156) | |
| Higher National Certificate or diploma | | | | | 4.251** | 0.003 | 4.205** | 0.003 |
| | | | | | (1.453 to 7.048) | | (1.409 to 7.001) | |
| University degree | | | | | 3.120 | 0.062 | 2.876 | 0.086 |
| | | | | | (-0.156 to 6.395) | | (-0.405 to 6.156) | |
| **Living situation** (reference: living alone) | | | | | | | | |
| With family | | | | | -7.143* | <0.001 | -7.132** | <0.001 |
| | | | | | (-10.108 to -4.178) | | (-10.095 to -4.168) | |
| With partner/spouse | | | | | -2.284* | 0.036 | -2.460* | 0.024 |
| | | | | | (-4.415 to -0.154) | | (-4.591 to -0.330) | |
| **Comorbidity** (Katz comorbidity index) | | | | | -0.930** | <0.001 | -0.936** | <0.001 |
| | | | | | (-1.428 to -0.433) | | (-1.432 to -0.439) | |

*(Continued)*

**Table 3.** (Continued)

| | Unadjusted models | | | | Adjusted models | | | |
|---|---|---|---|---|---|---|---|---|
| | **Model 1: unadjusted, no interaction** | | **Model 2: unadjusted, with interaction** | | **Model 3: adjusted, no interaction** | | **Model 4: adjusted, with interaction** | |
| **Constant** | 58.426*** | <0.001 | 58.388*** | <0.001 | 43.033*** | <0.001 | 43.083*** | <0.001 |
| | (56.020 to 60.831) | | (55.987 to 60.789) | | (39.018 to 47.048) | | (39.072 to 47.095) | |
| **Random effects parameters** | | | | | | | | |
| Patient-level (SD) | 11.690 | | 11.693 | | 8.099 | | 8.097 | |
| | (10.820–12.631) | | (10.824–12.631) | | (7.360 to 8.912) | | (7.360–8.909) | |
| Residual | 6.040 | | 5.990 | | 6.140 | | 6.099 | |
| | (5.637–6.472) | | (5.591–6.418) | | (5.719 to 6.592) | | (5.680–6.548) | |

AS/A Level: advanced subsidiary/advanced level; CI: confidence interval; EFI: electronic frailty index; GCSE: general certificate of education; GDS: geriatric depression scale; SD; standard deviation

*** p<0.001

** p < .001

* p<0.05

Another possible explanation is that severe frailty in itself is so debilitating for ADL that no moderating effects are observed.

Previously, Lohman et al. have shown that rapid increases in both frailty and depression predict nursing home admission and serious falls in community dwelling adults aged 51 years and over [23]. Furthermore, as with our analysis, when modelled together the effects of depression and frailty were attenuated suggesting that both frailty and depression explain vulnerability to adverse health outcomes. Indeed there is accumulating evidence to suggest that depressive symptoms contribute significantly to observed vulnerability to poor health in older people with frailty and that targeting depression might mitigate the negative effects of frailty on functioning and other health outcomes [24]. While the precise mechanisms that might explain such mediation are not yet clear, it is possible that disability in instrumental ADL is exacerbated in older people with frailty because depression is a disabling condition in older adults and is associated with increased number and severity of medical comorbidities and clusters of health-risk behaviours such as sedentary lifestyles [25, 26]. There is also the prospect that the relationship between depression and functioning is bi-directional. Low grip strength, a marker of physical functioning, is associated both cross-sectionally and longitudinally with depression [27]. Grip strength is highly correlated with upper body strength and predicts future ADLs [28]. Considered together, treating depression is likely to be an effective way to delay or reduce the likelihood of frailty progression in older adults.

## Implications for research and practice

There is a persuasive argument that the management of frailty can be optimised if it is conceptualised as a long-term condition. In this sense, frailty can be used to identify target groups of people with multimorbidity with complex needs such as those with combinations of mental and physical conditions and functional impairment [29]. Here, engaging proactive approaches that draw on the chronic care model and behaviour change interventions may have some utility [30]. While there is good evidence that integrated collaborative care interventions that target depression in people with long term conditions are effective, even in people with high levels of disability and multimorbidity, these approaches have yet to be proven effective in older adults with frailty [31, 32]. Similarly, there is evidence that behavioural activation for

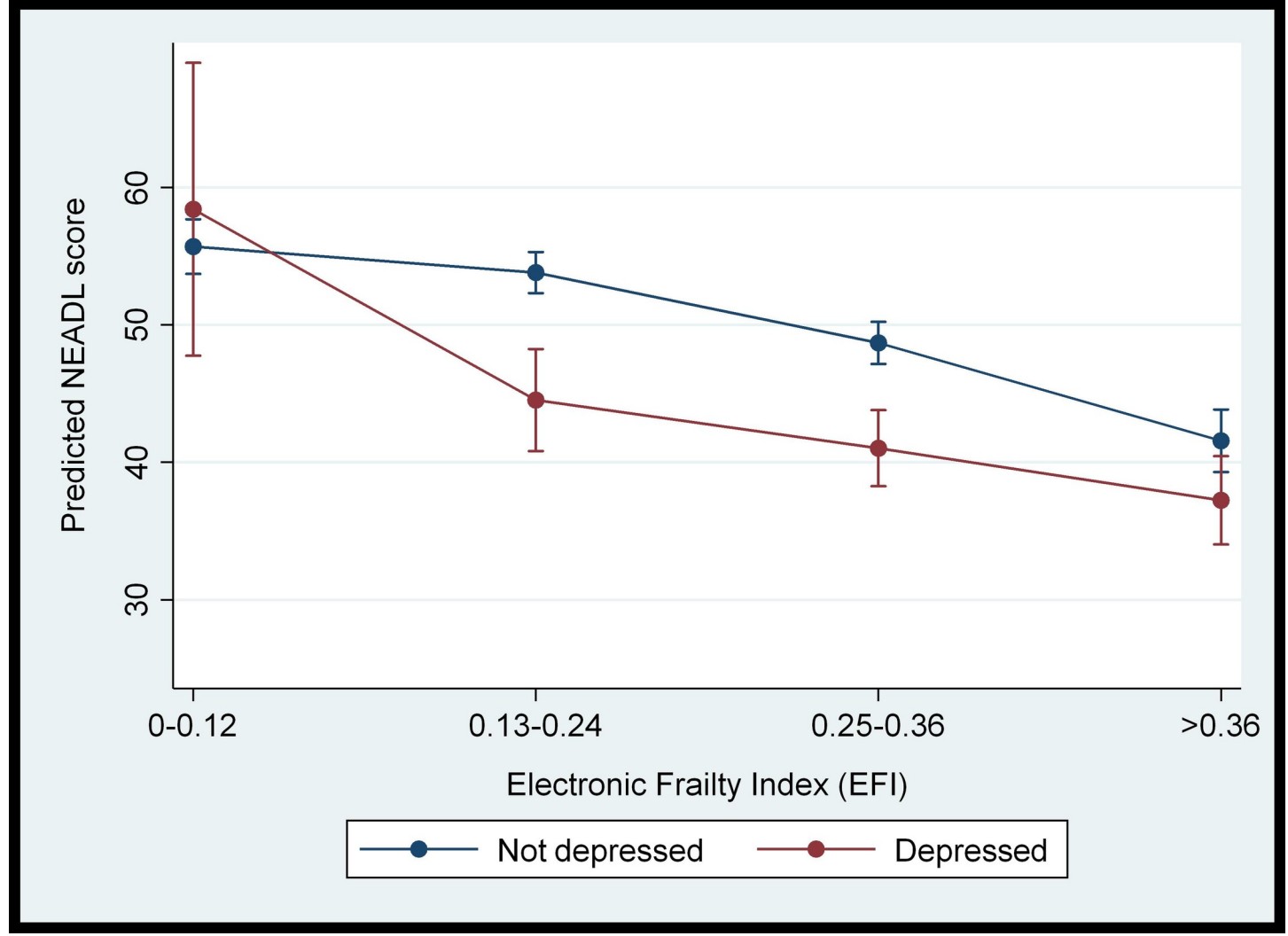

**Fig 2. Margins plot showing interaction between depression status (measured by GDS-15) and electronic frailty index in predicting NEADL score.**

depression in older adults is effective but trials to date have historically been small with significant methodological limitations [33]. Additionally, trials in behavioural activation have narrowly focused on depression rather than looking to impact both physical functioning and mental health outcomes in older adults.

There is also scope to go beyond traditional medical approaches and consider public health strategies that can delay or slow physical decline in older adults with frailty and depression. Here the natural environment as a community health asset could play a critical role in the management of frailty by making a positive impact on functioning and mental health. Green spaces in rural and urban areas have been shown in the general population to be highly beneficial to health and wellbeing and enhance social interaction [34]. There is also emerging evidence that exposure to the natural environment can confer health benefits in older adults. Higher coverage of urban green space is associated with reduced risk of all cause mortality, circulatory system-caused mortality, and stroke-caused mortality in community dwelling older adults aged ≥65 years [35]. Moreover, the frailty status of older adults living in neighbourhoods with higher levels of green space is more likely to improve over two years than in those

living in neighbourhoods with less green space [36]. The pathway to such frailty improvements might be through improving mental health. Recent analysis of the Whitehall II cohort has shown that higher residential greenness and proximity to any natural environment slows decline in walking speed and grip strength and this association is party mediated by mental health [37]. Nature based interventions can benefit both physical and mental health and could confer health benefits for older adults across the frailty trajectory. Well designed and robust studies are needed to test the effectiveness of public mental health interventions that can be mapped to frailty status and promote healthy ageing in older people.

## Strengths and limitations

A key strength of this study is the use of the CARE75+ cohort as a highly phenotyped primary care cohort with a recruitment rate of approximately 40%, comparable with other contemporary UK-based cohorts recruiting similar populations [38]. Findings are therefore likely to be representative of older adults recruited from primary care populations. However the sample was predominately white precluding exploration of whether ethnicity potentially explains variation in frailty and depression in older adults. Additionally, while the CARE75+ cohort includes an extensive range of health, social and economic outcome data it does not actively recruit older people from care homes, limiting opportunities to explore the impact of frailty and depression in older adults with the worst functioning. However, if people transitioned to a care home during the course of the study attempts are made to follow these people up. Furthermore, because we found that depression moderated the impact of frailty on instrumental ADL in only those with mild and moderate frailty it is likely that therapeutic interventions to manage depression in these groups will be delivered outside of care homes.

The CARE 75+ cohort study is an on-going study that started in January 2014 and not all participants in this study had reached the 6 month and 12 month follow-up assessment point at the time of the data extraction and analysis. As a result 214 participants were not included at 6-months, and an additional 117 participants were not included at 12 moths. These do not qualify as loss to follow-up but represent a lag in data accrual which is a feature of longitudinal studies. However we were able to collect sufficient outcomes on cases at follow-up to undertake suitably powered analyses. Other limitations relate to the instruments used to collect outcome data. The GDS-15 is not diagnostic but it does have proven capacity to screen for depressive symptoms across genders and age groups, from the young-old to the oldest-old [39]. Furthermore, the GDS has been shown to have predictive validity for mortality in populations aged $\geq$65 years with chronic heart failure, disability and cognitive impairments, suggesting it does have some utility in measuring depression in older adults [40]. However, there remains a debate about the performance of the GDS in people with cognitive impairment [41]. Depression and cognitive impairment often overlap. The study did not include a measure of cognitive impairment. Future studies should seek to clarify the independent contribution of depressive symptoms to outcome controlling for cognitive impairment. Finally, while grip strength is included in measures of frailty captured by CARE75+ we chose to use instrumental ADL as a proxy for physical functioning. Instrumental ADLs are essential to the maintenance of autonomy and independence and decrements in instrumental ADL are critical markers of progression of disability [42].

## Conclusion

We have shown using data from a large population-based cohort of older adults that frailty and depression independently predict disability in instrumental ADL. Depression moderated

the impact of frailty on instrumental ADL pointing to the potential for innovative solutions that target both physical and mental health in the management of frailty.

## Supporting information

**S1 Checklist. STROBE statement—checklist of items that should be included in reports of cohort studies.**
(DOCX)

## Author Contributions

**Conceptualization:** Peter A. Coventry, Andrew Clegg.

**Data curation:** Lesley Brown.

**Formal analysis:** Shehzad Ali.

**Funding acquisition:** Peter A. Coventry, Dean McMillan, Andrew Clegg, Simon Gilbody.

**Methodology:** Shehzad Ali.

**Writing – original draft:** Peter A. Coventry.

**Writing – review & editing:** Dean McMillan, Andrew Clegg, Lesley Brown, Christina van der Feltz-Cornelis, Simon Gilbody, Shehzad Ali.

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
