## [Decision Letter · Decision Letter 0]

13 May 2020

PONE-D-20-09338

Frailty and depression predict instrumental activities of daily living in older adults: a population-based longitudinal study using the CARE75+ cohort

PLOS ONE

Dear DR COVENTRY,

Thank you for submitting your manuscript to PLOS ONE. After careful consideration, we feel that it has merit but does not fully meet PLOS ONE’s publication criteria as it currently stands. Therefore, we invite you to submit a revised version of the manuscript that addresses the points raised during the review process.

We would appreciate receiving your revised manuscript by Jun 27 2020 11:59PM. To enhance the reproducibility of your results, we recommend that if applicable you deposit your laboratory protocols in protocols.io, where a protocol can be assigned its own identifier (DOI) such that it can be cited independently in the future. For instructions see: http://journals.plos.org/plosone/s/submission-guidelines#loc-laboratory-protocols

We look forward to receiving your revised manuscript.

Kind regards,

Pasquale Abete

Academic Editor

PLOS ONE

Journal Requirements:

"The funders had no role in study design, data collection and analysis, decision to

publish, or preparation of the manuscript."

5. Your ethics statement must appear in the Methods section of your manuscript. If your ethics statement is written in any section besides the Methods, please move it to the Methods section and delete it from any other section. Please also ensure that your ethics statement is included in your manuscript, as the ethics section of your online submission will not be published alongside your manuscript.

Additional Editor Comments (if provided):

According to Reviewer's comments the manuscripts needs a major revision.

Reviewers' comments:

Reviewer's Responses to Questions

**Comments to the Author**

1. Is the manuscript technically sound, and do the data support the conclusions?

Reviewer #1: Yes

Reviewer #2: Partly

Reviewer #3: Yes

2. Has the statistical analysis been performed appropriately and rigorously? 

Reviewer #1: Yes

Reviewer #2: No

Reviewer #3: Yes

3. Have the authors made all data underlying the findings in their manuscript fully available?

Reviewer #1: Yes

Reviewer #2: No

Reviewer #3: Yes

4. Is the manuscript presented in an intelligible fashion and written in standard English?

Reviewer #1: No

Reviewer #2: Yes

Reviewer #3: Yes

5. Review Comments to the Author

Reviewer #1: The authors evaluate if depression contributes, independently and/or in interaction with frailty, to loss of independence in instrumental activities of daily living (ADL) in older adults with frailty. They studied Longitudinal cohort study of people aged ≥75 years living in the community by usying multi-level linear regression model to quantify the relationship between depression (≥5 Geriatric Depression Scale) and frailty (electronic frailty index), and instrumental activities of daily living (Nottingham Extended Activities of Daily Living scale; range: 0-66; higher score implies greater independence). They included 553 participants at baseline; 53% were female with a mean age of 81 (5.0 SD) years. Depression and frailty (moderate and severe levels) were independently associated with reduced instrumental activities of daily living scores. Moreover, depression interacted with frailty to further reduce instrumental activities of daily living score in individuals with mild or moderate frailty. These relationships remained significant after adjusting for confounders.

The manuscript is interesting. However I have some concerns about the number of patients. What about the sample size and the power of the study? What about the comorbidities such as heart failure? Please see and discuss Liguori I et al. Depression and chronic heart failure in the elderly: an intriguing relationship. J Geriatr Cardiol. 2018 Jun;15(6):451-459.

Reviewer #2: The data has to be requested from the data controller's website. Technically the authors cannot make it fully available.

The authors used a subsample of participants in the CARE75+ cohort to examine the association between depression, frailty and instrumental activities of daily living (IADL) from baseline to 12 months of follow-up. The analytic approach is a linear mixed model with random intercept. The presentation and presentation of results can be improved. Answers to the following questions can improve a reader’s understanding of the validity of the results and conclusions.

1). Based on the flow chart, 106 participants were excluded from the study due to “delay between participant consent and baseline data being uploaded.” Were their baseline data available? If not, they probably would have been counted as “lost to study entry.” So my guess is their data were available. Then, what is the justification for this exclusion?

2). There was substantial amount of attrition (39% at 6 months and 60% at 6 months). It is known that linear mixed effects models are consistent under the assumption of missing at random. However, given this cohort’s age, it is unlikely the attrition was at random. The authors were not clear how many died, transitioned to a care home and was not followed, and was lost to follow-up due to other reasons. When attrition is not at random, the mixed effects model estimates are biased.

3). After estimating a linear mixed model, the authors presented marginal effects in Figure 2. Given that the analysis was done in Stata, I assume the authors used the margins statement after the linear mixed model for the marginal effects. However, the margins command after a mixed model in Stata only calculates the fixed effects part of the model, thus is not a real marginalization. Granted that if the missing at random assumption holds, the fixed part of the mixed model is equivalent to the population-averaged effects; however, given 2) this is unlikely to be true. The authors could take a look at the article by Rouanet et al. (2019) “Interpretation of mixed models and marginal models with cohort attrition due to death and drop-out” in Statistical Methods in Medical Research.

4). The authors speculated that “the absence of an effect in those with severe frailty stem from an underpowered analysis owing to smaller numbers.” However, the authors did not present the number of people with depression and severe frailty in each time period. A tabulate of sample size over time will be helpful.

Reviewer #3: The authors in this study evaluate if depression contribute, independently and/or in interaction with frailty, to loss of independence in instrumental activities of daily living (IADL) in older adults with frailty in a longitudinal cohort study of people aged ≥75 years living in the community by using multi-level linear regression model. They included 553 participants at baseline. Depression and frailty (moderate and severe levels) were independently associated with reduced instrumental activities of daily living scores. Moreover, depression interacted with frailty to further reduce instrumental activities of daily living score in individuals with mild or moderate frailty. These relationships remained significant after adjusting for confounders.

The manuscript is very interesting. However I have some concerns about adjusting confounders: given that cognitive impairment is an independent factor for ADL and IADL loss and that depression and dementia are often overlapping in the elderly population, has cognitive impairment been evaluated in the sample? Furthermore, GDS is not a validated tool for the evaluation of depression in patients with cognitive impairment. Indeed I think that some clarifications are necessary about effects of depression symptoms on comorbidities, please see and discuss “Depressive symptoms predict mortality in elderly subjects with chronic heart failure. Testa G, Cacciatore F, Galizia G, Della-Morte D, Mazzella F, Gargiulo G, Langellotto A, Raucci C, Ferrara N, Rengo F, Abete P. Eur J Clin Invest. 2011 Dec;41(12):1310-7”.

6. PLOS authors have the option to publish the peer review history of their article (what does this mean?). If published, this will include your full peer review and any attached files.

Reviewer #1: No

Reviewer #2: No

Reviewer #3: No

---

## [Author Response · Author response to Decision Letter 0]

10 Sep 2020

We have uploaded a response to the reviewers table as part of our revised submission.

As requested, in the cover letter, we have addressed the issue about data availability and given details about how requests for data access can be made.

---

## [Decision Letter · Decision Letter 1]

23 Sep 2020

PONE-D-20-09338R1

Frailty and depression predict instrumental activities of daily living in older adults: a population-based longitudinal study using the CARE75+ cohort

PLOS ONE

Dear Dr. COVENTRY,

Thank you for submitting your manuscript to PLOS ONE. After careful consideration, we feel that it has merit but does not fully meet PLOS ONE’s publication criteria as it currently stands. Therefore, we invite you to submit a revised version of the manuscript that addresses the points raised during the review process.

We look forward to receiving your revised manuscript.

Kind regards,

Pasquale Abete

Academic Editor

PLOS ONE

Additional Editor Comments (if provided):

According to Reviewer's comments the manuscript needs a major revision.

Sincerely,

P. Abete

Reviewers' comments:

Reviewer's Responses to Questions

**Comments to the Author**

1. If the authors have adequately addressed your comments raised in a previous round of review and you feel that this manuscript is now acceptable for publication, you may indicate that here to bypass the “Comments to the Author” section, enter your conflict of interest statement in the “Confidential to Editor” section, and submit your "Accept" recommendation.

Reviewer #1: All comments have been addressed

Reviewer #2: (No Response)

Reviewer #3: All comments have been addressed

2. Is the manuscript technically sound, and do the data support the conclusions?

Reviewer #1: Yes

Reviewer #2: Partly

Reviewer #3: Yes

3. Has the statistical analysis been performed appropriately and rigorously? 

Reviewer #1: Yes

Reviewer #2: No

Reviewer #3: Yes

4. Have the authors made all data underlying the findings in their manuscript fully available?

Reviewer #1: Yes

Reviewer #2: No

Reviewer #3: Yes

5. Is the manuscript presented in an intelligible fashion and written in standard English?

Reviewer #1: Yes

Reviewer #2: Yes

Reviewer #3: Yes

6. Review Comments to the Author

Reviewer #1: All comments have been addressed. The manuscript presented in an intelligible fashion and written in standard English. No further comments.

Reviewer #2: The authors responded to my first question by changing the description of why the 106 patients were excluded.

The authors responded to my second question by saying “The study recorded a total of 14 deaths; half of these participants, i.e. 7/14, completed all three waves of the study (i.e. they died after the 12-month follow-up period).” In the main text this information should be given. In the follow chart and in the main text, the reason for loss to follow up (not death) should be given, at six months 214 (7 died) patients were not included ; and at 12 months 331 (7 died) patients were not included; what are the reason for the exclusions?

The model for 12-month loss to follow up should not be among only those patients with non-missing 6-month data, it should be among all 553 patients. The authors should show in their response letter the logistic regression for loss to follow up at 6 months with age and baseline frailty and depression as predictors, although I do not believe this necessarily mean anything because we still don’t know if other variables are related to loss to follow up. Descriptive statistics of all variables in Table 1 should be given for those with and those without loss to follow up, one table for 6-month; one table for 12-month. We can from the descriptive statistics get a glimpse of how different the two groups are.

The authors added Table 2 (cross-tabulation of depression and frailty at baseline, 6 months and 12 months) in response to my question about their speculation that “the absence of an effect in those with severe frailty stem from an underpowered analysis owing to smaller numbers”, we can see that there are only 2 patients with depression and no frailty in 12 months. This fact makes the models with interaction between depression and frailty very questionable. No descriptive data were given for the outcome NEAL at baseline, 6 months or 12 months. One cannot get a sense of how big the effects (coefficients) are relative to the distribution of the outcome.

Reviewer #3: All comments has been addressed. The manuscript is improved and is suitable for pubblication in present form

7. PLOS authors have the option to publish the peer review history of their article (what does this mean?). If published, this will include your full peer review and any attached files.

Reviewer #1: No

Reviewer #2: No

Reviewer #3: No

---

## [Author Response · Author response to Decision Letter 1]

10 Nov 2020

We have compiled a response to reviewer comments table that includes author responses to reviewer number 2. These responses are in a table that we have included in this revised submission.

---

## [Decision Letter · Decision Letter 2]

2 Dec 2020

Frailty and depression predict instrumental activities of daily living in older adults: a population-based longitudinal study using the CARE75+ cohort

PONE-D-20-09338R2

Dear Dr. COVENTRY,

We’re pleased to inform you that your manuscript has been judged scientifically suitable for publication and will be formally accepted for publication once it meets all outstanding technical requirements.

Kind regards,

Pasquale Abete

Academic Editor

PLOS ONE

Additional Editor Comments (optional):

No further comments.

Reviewers' comments:

Reviewer's Responses to Questions

**Comments to the Author**

1. If the authors have adequately addressed your comments raised in a previous round of review and you feel that this manuscript is now acceptable for publication, you may indicate that here to bypass the “Comments to the Author” section, enter your conflict of interest statement in the “Confidential to Editor” section, and submit your "Accept" recommendation.

Reviewer #2: All comments have been addressed

2. Is the manuscript technically sound, and do the data support the conclusions?

Reviewer #2: Yes

3. Has the statistical analysis been performed appropriately and rigorously? 

Reviewer #2: Yes

4. Have the authors made all data underlying the findings in their manuscript fully available?

Reviewer #2: No

5. Is the manuscript presented in an intelligible fashion and written in standard English?

Reviewer #2: Yes

6. Review Comments to the Author

Reviewer #2: (No Response)

7. PLOS authors have the option to publish the peer review history of their article (what does this mean?). If published, this will include your full peer review and any attached files.

Reviewer #2: No

---

## [Editor Report · Acceptance letter]

4 Dec 2020

PONE-D-20-09338R2 

Frailty and depression predict instrumental activities of daily living in older adults: a population-based longitudinal study using the CARE75+ cohort 

Dear Dr. Coventry:

I'm pleased to inform you that your manuscript has been deemed suitable for publication in PLOS ONE. Congratulations! Your manuscript is now with our production department. 

Kind regards, 

on behalf of

Prof. Pasquale Abete 

Academic Editor

PLOS ONE